# An Unsupervised Approach for Artifact Severity Scoring in Multi-Contrast MR Images

**Savannah P. Hays**[1] iD                                          SHAYS6@JHU.EDU
[1] *Department of Electrical and Computer Engineering, Johns Hopkins University, USA*

**Lianrui Zuo**[2] iD                                          LIANRUI.ZUO@VANDERBILT.EDU
[2] *Department of Electrical and Computer Engineering, Vanderbilt University, USA*

**Blake E. Dewey**[3] iD                                          BLAKE.DEWEY@JHU.EDU
[3] *Department of Neurology, Johns Hopkins School of Medicine, USA*

**Samuel W. Remedios**[4] iD                                          SREMEDI1@JHU.EDU
[4] *Department of Computer Science, Johns Hopkins University, USA*

**Jinwei Zhang**[1]                                          JWZHANG@JHU.EDU
**Ellen M. Mowry**[3] iD                                          EMOWRY1@JHMI.EDU
**Scott D. Newsome**[3] iD                                          SNEWSOM2@JHMI.EDU
**Aaron Carass**[1] iD                                          AARON_CARASS@JHU.EDU
**Jerry L. Prince**[1,4] iD                                          PRINCE@JHU.EDU

**Editors:** Under Review for MIDL 2025

## Abstract

Quality assurance (QA) in magnetic resonance (MR) imaging is critical but remains a challenging and time-intensive process, particularly when working with large-scale, multi-site imaging datasets. Manual QA methods are subjective, prone to inter-rater variability, and impractical for high-throughput workflows. Existing automated QA methods often lack generalizability to diverse datasets or fail to provide interpretable insights into the causes of poor image quality. To address these limitations, we introduce an unsupervised and interpretable QA framework for multi-contrast MR images that quantifies artifact severity. By assigning a numerical score to each image, our method enables objective, consistent evaluation of image quality and highlights specific levels of artifact presence that can impair downstream analysis. Our framework employs an unsupervised contrastive learning approach, leveraging simulated artifact transformations, including random bias, noise, anisotropy, and ghosting, to train the model without requiring manual labels or preprocessing. A margin-based contrastive loss further enables differentiation between varying levels of artifact severity. We validate our framework using simulated artifacts on a public dataset and real artifacts on a private clinical dataset, demonstrating its robustness and generalizability for automatic MR image QA. By efficiently evaluating image quality and identifying artifacts prior to data processing, our approach streamlines QA workflows and enhances the reliability of subsequent analyses in both research and clinical settings.

**Keywords:** MRI, Quality Assurance, Artifact Detection

## 1. Introduction

Magnetic resonance (MR) imaging is a cornerstone of medical diagnostics and research, offering unparalleled insights into the structure and function of tissues (Bernstein et al., 2004). However, the quality of MR images can be significantly compromised by various artifacts, including bias field inhomogeneities, noise, motion, anisotropic resolution, and ghosting (Zaitsev et al., 2015). These artifacts not only degrade image interpretability but also affect downstream analyses, potentially leading to erroneous conclusions in both clinical and research settings (Zaitsev et al., 2015). Consequently, robust quality assurance (QA) of MR images is essential to ensure the reliability of data and analyses.

Artifacts in MR images can have far-reaching implications. For instance, in clinical settings, poor image quality can lead to misdiagnoses or necessitate costly and time-consuming rescans. In research, artifacts can bias analyses, particularly in multi-site studies where variability in image quality is compounded by differences in scanner hardware, acquisition protocols, and patient populations. Advanced techniques are particularly vulnerable to artifact-induced errors, underscoring the need for automated QA systems.

Existing MR image QA methods often involve manual inspection, which is labor-intensive, subjective, and prone to inter-rater variability (Esteban et al., 2017; Alfaro-Almagro et al., 2018). Automated methods frequently rely on preprocessing steps such as image registration or supervised training paradigms, which require large, labeled datasets and are susceptible to biases inherent in the training data (Esteban et al., 2017). Traditional quality control (QC) metrics like signal-to-noise ratio (SNR) and contrast-to-noise ratio (CNR) are limited in their ability to capture complex artifacts. Moreover, these approaches fail to provide interpretable outputs that can guide users in identifying corrupted images within their dataset.

Recent advances in deep learning have led to the development of supervised artifact detection models, which rely on labeled training data to classify images as high- or low-quality. For example, Oksuz (2021) introduced a convolutional neural network (CNN)-based approach for detecting and correcting motion artifacts in brain MRI, demonstrating the effectiveness of deep learning in artifact detection. However, supervised models require large annotated datasets and often struggle to generalize to unseen artifact types.

More recently, self-supervised and transfer learning methods have been explored for MRI quality assessment. Vakli et al. (2023) proposed an end-to-end deep learning model trained on image quality metrics to classify motion artifacts. Similarly, Loizillon et al. (2024) leveraged simulated artifacts and transfer learning to develop an automated MRI QA framework. While supervised approaches have shown promise in MR image quality assessment, they rely heavily on large, manually labeled datasets, which can be challenging to acquire and prone to inter-rater variability. In particular, some artifacts—such as subtle anisotropic blurring or low-level bias field distortions—are not always easily recognizable by human reviewers, making manual annotation subjective and inconsistent. Additionally, supervised models are inherently limited by the quality and diversity of their training data, often struggling to generalize to unseen artifacts or new imaging protocols across different scanners.

In contrast, an unsupervised approach circumvents these challenges by learning a continuous artifact severity scale without requiring explicit labels. This allows for more scalable training across large, heterogeneous datasets while avoiding biases introduced by expert

annotations. Moreover, unsupervised learning is particularly beneficial for multi-site studies where variations in scanner hardware, acquisition parameters, and patient populations introduce unpredictable artifact distributions. Our work builds upon these methods by employing a fully unsupervised approach that does not require explicit artifact labels, making it more flexible and scalable to diverse datasets.

MRIQC (Esteban et al., 2017) is a widely used tool that reports a range of image quality metrics (IQMs) for structural, functional, and diffusion-weighted MR images. For structural images, it supports both $T_1$-weighted ($T_1$-w) and $T_2$-weighted ($T_2$-w) modalities and provides quantitative assessments based on noise, entropy, and contrast-related measurements. While MRIQC generates detailed reports, interpreting these IQMs can be challenging, especially for large datasets. It remains difficult to establish consistent thresholds and optimal combinations of IQMs to determine whether an image should pass or fail QA.

Among MRIQC's structural IQMs, we identified four key metrics that are relevant to our work: coefficient of joint variation (CJV), contrast-to-noise ratio (CNR), entropy focus criterion (EFC), and foreground-background energy ratio (FBER). CJV (Ganzetti et al., 2016) measures intensity variability between gray matter (GM) and white matter (WM) and is sensitive to head motion and intensity non-uniformity artifacts. CNR (Magnotta and Friedman, 2006) assesses how well the intensity distributions of GM and WM are separated. EFC (Atkinson et al., 1997) is based on Shannon entropy and quantifies the amount of ghosting and blurring induced by motion artifacts. FBER (Shehzad et al., 2015) measures the mean energy of voxel intensities within the brain relative to areas outside the brain.

Although MRIQC provides useful quantitative measures, determining a single threshold or combination of IQMs that universally defines poor-quality images remains a challenge. The complexity of these metrics and their dataset dependence further motivate the need for a more interpretable and automated QA framework.

In this work, we propose an automatic method for QA of MR images that addresses these limitations. Our approach generates objective artifact severity scores without relying on subjective expert labels. This enables a more scalable and generalizable solution for automated MR image QA, particularly in multi-site studies where image quality can vary significantly due to differences in scanner hardware and acquisition settings. By learning a continuous severity scale rather than classifying images into discrete quality categories, our model is capable of capturing subtle variations in artifact intensity that may not be readily discernible through manual review. These scores can be used to set adaptive thresholds for automatic QA, enabling users to systematically identify and exclude poor-quality images prior to downstream processing. Our method does not require image preprocessing steps, making it computationally efficient and broadly applicable across different datasets. It can be directly used on NifTI files of many MR image modalities.

Central to our approach is an unsupervised training framework based on contrastive learning inspired by that of Zuo et al. (2023a,b). By leveraging a data loader that applies a diverse set of realistic transformations—including random bias, noise, anisotropy, and ghosting—we simulate a wide range of artifact severities. This enables the model to learn discriminative features that correlate with image quality without the need for labeled training data. Contrastive learning, which pulls similar samples closer in feature space while pushing dissimilar ones apart, is particularly well-suited for this task as it learns a meaningful ranking of artifact severity without explicit supervision. Unlike classification-based approaches that

rely on pre-defined labels, contrastive learning naturally organizes images into a continuous quality spectrum, making it more adaptable to real-world variations in artifact intensity. Furthermore, since some artifacts can be difficult to detect through traditional quality metrics, a contrastive framework allows the model to learn subtle but clinically relevant degradation patterns that may not be captured by predefined quality scores. Furthermore, our method is capable of recognizing and penalizing poor resolution, a critical but often overlooked aspect of image quality.

Our approach is designed to address the growing reliance on large, multi-site datasets where artifact heterogeneity poses significant challenges. By providing interpretable artifact severity scores, our method promotes reproducibility and reliability in medical imaging studies. Moreover, it can be seamlessly integrated into automated pipelines for preprocessing large-scale datasets, significantly reducing the burden of manual QC and enabling more efficient use of resources.

We validate our approach by comparing it with MRIQC (Esteban et al., 2017), demonstrating its effectiveness in accurately identifying low-quality images and providing actionable insights for improving data quality. Our method offers a scalable and interpretable solution for automatic MR image QA, paving the way for more reliable and reproducible analyses in medical imaging. Our model is open source and is publicly available from: https://github.com/shays15/artifact_scoring.

## 2. Methods

### 2.1. Overview

Our model assigns an artifact score to an MR image, quantifying the level of artifacts present. The model was trained in an unsupervised fashion using a diverse set of simulated artifacts. It leverages triplet loss and the L2 loss to learn meaningful representations of image quality. The overall training and inference workflow is illustrated in Fig. 1.

### 2.2. Training Dataset

We trained our model using 297 structural MR volumes from the TRaditional vs. Early Aggressive Therapy for Multiple Sclerosis (TREAT-MS) pragmatic, clinical trial (NCT03500328). These scans were acquired from seven different imaging sites and included multiple contrasts: $T_1$-w, $T_2$-w, $T_2$-w FLAIR, and proton density (PD) images. To improve generalization across scanners and imaging conditions, only high-quality images were included in the training dataset. Prior to training, the images were N4 bias field corrected (Tustison et al., 2010) to address intensity inhomogeneities and 2D acquisitions were super-resolved (Remedios et al., 2023) to ensure consistent resolution. These preprocessing steps were necessary for artifact simulation but are not required during inference, ensuring the model remains applicable to diverse datasets without additional preprocessing.

### 2.3. Model Architecture

Our model operates on 2D MR image slices, making it computationally efficient while allowing for a larger number of training samples. Our model takes any input and resizes it to 224×224. The architecture consists of two key components: a custom convolutional block

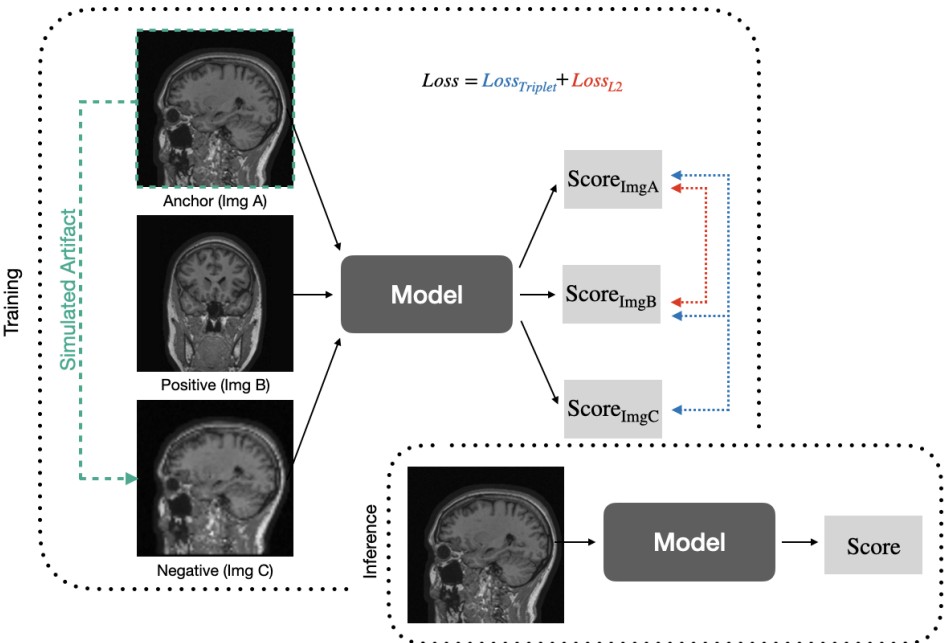

Figure 1: Training and inference workflow for our model. During training, three images are used to calculate the total loss. Img A and Img B are two different, clean image slices, while Img C is Img A with a randomly simulated artifact.

and a primary encoder. The convolutional block is composed of two convolutional layers with $3 \times 3$ kernels, each followed by instance normalization and a LeakyReLU activation function, allowing the network to effectively capture spatial features. The encoder stacks two of the custom convolutional blocks with increasing channel dimensions, followed by a single convolutional layer to reduce the feature dimensions and an adaptive average pooling operation to reduce the output to the desired dimension. An absolute value function is applied to enforce non-negative scores for interpretability. During inference, an MR volume is assigned a single score by averaging scores across the middle 60% of slices.

## 2.4. Training and Artifact Simulation

To effectively train the model, we developed a data augmentation module that simulates common MR image artifacts in a controlled manner. These artifact transformations were implemented using the TorchIO library (Pérez-García et al., 2021) and include random noise, random ghosting, random bias field, and random anisotropy. Random noise introduces varying levels of noise in the images. Random ghosting simulates motion-induced ghosting with varying intensities and repetition. Random bias field introduces intensity inhomogeneities mimicking scanner-specific artifacts. Random anisotropy reduces spatial resolution to simulate anisotropic acquisitions. Each transformation is parameterized and assigned a calculated severity score (SS) in the range $[0, 1]$, as detailed in Table 1. The parameters are randomly sampled to ensure continuity in the artifact space exposing the

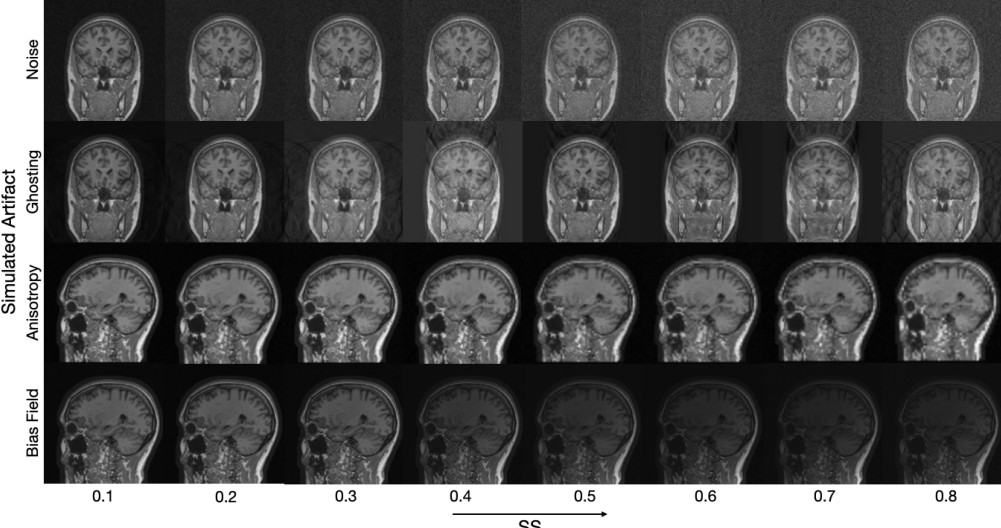

Figure 2: Increasing severity scores (SS) in the range $[0, 1]$ (left to right) of the simulated artifacts (from top to bottom: noise, ghosting, anisotropy, bias) seen during training.

Table 1: Each artifact and its parameters used in the severity score (SS). The parameters were uniformly sampled in the corresponding range to ensure continuity of the artifact space.

| Artifact | Input | Parameters | Severity Score (SS) |
|----------|-------|------------|---------------------|
| **Noise** | `std` | $\mathcal{U}[0.005, 0.2]$ | $\dfrac{\text{std} - 0.005}{0.2 - 0.005}$ |
| **Ghosting** | `num_ghosts` 
 `intensity` | $\mathcal{U}\{2, \cdots, 10\}$ 
 $\mathcal{U}[0.2, 1.5]$ | $\dfrac{(\text{intensity} - 0.2) + \frac{\text{num\_ghosts}}{10}}{(1.5 - 0.2) + 1}$ |
| **Bias Field** | `coefficients` | $\mathcal{U}[0.01, 0.3]$ | $\dfrac{\text{coefficients} - 0.01}{0.3 - 0.01}$ |
| **Anisotropy** | `scale` | $\mathcal{U}[1, 4]$ | $\dfrac{\text{scale} - 1}{4 - 1}$ |

model to diverse image degradations. Figure 2 illustrates example images with increasing severity scores. During training, the severity scores are used as the margin in the triplet loss to emphasize the relative differences between clean and artifact-degraded images. The model itself is only constrained to produce non-negative outputs. It is not constrained to produce outputs within a specific range. This design allows the model to flexibly assign scores based on the learned features.

## 2.5. Loss Functions

During training, three images are passed through the network: a clean (anchor) image, a second clean image (positive sample), and the anchor image with a simulated artifact (negative sample). To learn robust artifact representations, the model is optimized using two loss functions: L2 loss and triplet loss.

**L2 Loss:** The L2 loss ensures that clean images are consistently assigned low severity scores, encouraging stability in the artifact-free case:

$$\mathcal{L}_{L2} = \frac{1}{N} \sum_{i=1}^{N} (S_i - S_j)^2 \tag{1}$$

where $S_i$ and $S_j$ are the predicted severity scores of two clean images.

**Triplet Loss:** The triplet loss enforces a ranking such that the clean anchor image receives a lower severity score than the artifact-degraded negative image:

$$\mathcal{L}_{\text{triplet}} = \sum_{i=1}^{N} \max(0, S_i^{\text{anchor}} - S_i^{\text{positive}} + m) + \max(0, S_i^{\text{negative}} - S_i^{\text{anchor}} + m) \tag{2}$$

where $m$ is a dynamic margin based on artifact severity.

While triplet loss ensures relative ranking between clean and artifact images, it does not enforce absolute scale consistency. L2 loss stabilizes the training by anchoring clean images to low scores, preventing score drift. The margin $m$ in the triplet loss adapts dynamically based on artifact severity. For high-severity artifacts, $m$ is larger, ensuring clearer separability between clean and degraded images. This approach acts as a form of hard negative mining, where the model prioritizes distinguishing the most challenging cases.

## 3. Experiments and Results

### 3.1. Public Dataset

We first evaluated our model on a sample from the OASIS dataset (Marcus et al., 2007) ($N = 20$). Based on our model scores during inference and visual interpretation, a value of 1 is a reasonable threshold for the model scoring. Anything with a score under 1 is considered to be of sufficient quality for subsequent processing. Anything with a higher score than 1 should be excluded from processing or undergo correction steps dependent on the artifact type. To assess the model's ability to rank artifact severity, we simulated artifacts on the OASIS images resulting in evaluation of 120 images. We compared our model's scores with MRIQC (v25.0.0) IQMs (Table 2). Our model consistently ranked images by artifact severity, whereas MRIQC's IQMs showed inconsistent trends across different artifacts. Notably, MRIQC processing failed to report IQMs on images of $T_2$-w contrast and high levels of simulated artifacts. This restricted our experiments to one modality and one artifact type and SS rather than testing a variety of severities per artifact. We performed all computations on a system with a 16-core processor running at 3.22 GHz per core and 251.66 GB of RAM. On successful cases, MRIQC took between $7 - 9$ minutes per volume. Our model took about 1 second per volume.

Table 2: The specified artifact type is added with the specified severity score (SS) as outlined in Table 1. We compare the MRIQC statistics against the score from our model ($N = 20$ for each artifact type). The arrows should indicate improving image quality. We report the Pearson correlation coefficient between the SS and each of the MRIQC IQMs and our model score in the last row of the table.

| Artifact Type | SS | MRIQC | | | | Ours ↓ |
|---|---|---|---|---|---|---|
| | | CJV ↓ | CNR ↑ | EFC ↓ | FBER ↑ | |
| None | 0.0 | 0.77±0.13 | 1.14±0.25 | 0.49±0.05 | 6962±2097 | 0.17±0.47 |
| Bias | 0.1 | 0.80±0.19 | 1.08±27 | 0.51±0.60 | 2367±1572 | 0.01±0.03 |
| Motion | 0.3 | 0.87±0.19 | 1.05±0.27 | 0.51±0.05 | 5501±2010 | 1.74±0.49 |
| Anisotropy | 0.6 | 0.78±0.05 | 1.32±0.23 | 0.52±0.05 | 9876±3326 | 2.32±0.48 |
| Ghosting | 0.8 | 1.03±0.18 | 0.82±0.12 | 0.53±0.06 | 4161±1852 | 2.34±0.60 |
| Noise | 0.9 | – | – | – | – | 3.54±0.06 |
| **Pearson $r$** | | 0.36 | -0.16 | 0.13 | 0.35 | 0.92 |

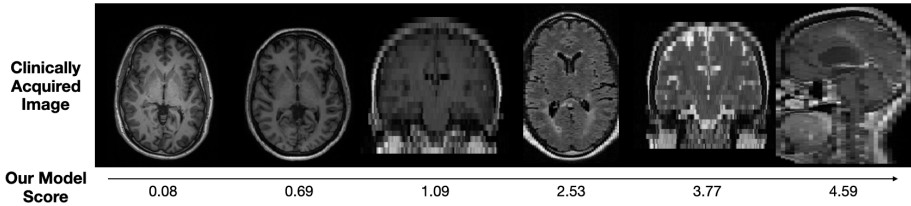

Figure 3: Clinically acquired images from the TREAT-MS dataset. Although images are acquired at various sites following a standardized protocol, several low resolution 2D acquisitions are observed.

### 3.2. Private Clinical Dataset

We further validated our model using 124 structural MR volumes from the TREAT-MS pragmatic, clinical trial (NCT03500328). These images were acquired following a standardized protocol but still exhibited substantial variation in image quality, particularly in resolution. Many images are 2D acquired at varying levels of resolutions. Figure 3 shows clinically acquired images ranked by their scores. Our model assigned scores close to 0 for high-quality images and higher scores for low-resolution 2D acquisitions, indicating strong detection of anisotropic resolution artifacts. Based on our findings, we recommend images with scores $\leq 1$ are of sufficient quality for downstream processing. Images with scores $> 1$ should be flagged for review or undergo artifact correction (e.g., super-resolution, background removal). 83% of this private clinical dataset scored $> 1$. This was expected as the majority of the images are 2D clinically acquired with thick slice thicknesses. This evaluation demonstrates that our model effectively identifies artifact severity in both public and clinical datasets, providing interpretable scores that facilitate automated MR image quality assurance.

## 4. Discussion and Conclusion

In this study, we introduced an unsupervised, interpretable framework for QA in multi-contrast MR imaging. By training on diverse data with simulated artifact transformations, our model produces scores that correlate with image quality, offering an efficient and scalable solution for QA workflows. Our approach measures apparent image quality rather than anatomical realness. Unlike existing methods, our approach eliminates the need for manual labels or preprocessing. The framework is compatible with $T_1$-w, $T_2$-w, $T_2$-w FLAIR, and PD images. Although MRIQC can be directly run using their provided docker, we experienced many issues and limitations running this program. MRIQC requires data to be in BIDS format, this implies some level of preprocessing which we wish to avoid. For structural images, it can only retrieve IQMs for $T_1$-w and $T_2$-w images, while our approach can handle $T_1$-w, $T_2$-w, FLAIR, and PD images. However, MRIQC can also handle diffusion data which we cannot currently. There is also a large variation in computational time between the two methods. For many of the $T_2$-w images and images with a simulated artifact, MRIQC failed to complete—which we found was a common experience in the MRIQC discussion forums with no known solution. This is the primary reason for our limited experiment and missing metrics for the noise artifact type in Table 2. We did not observe much slice variation in the scoring of our model. We credit this to only using the middle 60% of slices. A key feature of the framework is the threshold for artifact scores. Based on empirical results, a threshold of 1 was determined as a reasonable cutoff for distinguishing between high- and low-quality images. During training, the margin in the triplet loss is derived from simulated artifact severity scores bounded between 0 and 1. This margin emphasizes the distinction between clean and artifact-ridden images while allowing flexibility for the model's outputs during inference. Consequently, the model's unrestricted scoring capability results in a broader range of scores (0 to 6) during real-world application. The simulated artifacts used in training currently focus on one artifact type at a time. In practice, MR images can exhibit multiple overlapping artifacts. We hypothesize that the model detecting multiple artifacts might be a reasoning for the broader range of scores during application. Future work would incorporate combinations of simulated artifacts during training, making the model more robust and reflective of real-world data. Additionally, incorporating artifact-specific weighting into the scoring process could improve interpretability by assigning higher penalties to artifacts that are particularly detrimental to downstream analysis.

The proposed framework has significant implications for large-scale studies, where it can streamline data preprocessing by automating QA and identifying low-quality images prior to analysis. Its unsupervised nature ensures adaptability across different datasets without reliance on labeled training data. This adaptability makes it particularly valuable for multi-site studies with heterogeneous imaging protocols and scanner hardware. Overall, this work lays a robust foundation for advancing automated MR image QA and emphasizes the importance of interpretable, unsupervised learning solutions in medical imaging. By providing artifact severity scores that are both flexible and actionable, this framework enables more efficient and reliable analyses, paving the way for enhanced QA workflows in both clinical and research settings.

## Acknowledgments

This material is partially supported by the Johns Hopkins University Percy Pierre Fellowship (Hays) and the National Science Foundation Graduate Research Fellowship under Grant No. DGE-2139757 (Hays) and Grant No. DGE-1746891 (Remedios). Development is partially supported by FG-2008-36966 (Dewey), CDMRP W81XWH2010912 (Prince), NIH R01 CA253923 (Landman), NIH R01 CA275015 (Landman), the National MS Society grant RG-1507-05243 (Pham) and Patient-Centered Outcomes Research Institute (PCORI) grant MS-1610-37115 (Newsome and Mowry). The statements in this publication are solely the responsibility of the authors and do not necessarily represent the views of the Patient-Centered Outcomes Research Institute (PCORI), its Board of Governors or Methodology Committee.

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
