# OpenReview forum: "An Unsupervised Approach for Artifact Severity Scoring in Multi-Contrast MR Images"
_MIDL.io/2025/Conference — MIDL 2025 Poster_

### Official Review · Reviewer_8Z4Q · 2025-02-18

**Confidence:** 3
**Preliminary Rating:** 3
**Recommendation:** Poster
**Final Rating:** 3

**Summary:**

The authors present a contrastive learning-based method to score artifact severity in MR images of different contrasts. Their method is compared to the widely used tool MRIQC, and the results suggest that the proposed approach captures the severity of artifacts more reliably and in a more gradual manner. This contribution has the potential to benefit the neuroimaging field, especially in the context of large-scale multi-site studies.

**Strengths:**

- The proposed method addresses an important issue in neuroimaging, particularly in quality control for large-scale datasets.
- The paper is generally well-written and well-motivated, with a clear presentation of the problem and the need for a better artifact severity scoring approach.
- The method seems to provide a more interpretable and gradual assessment of artifacts compared to MRIQC, which could be valuable in practical applications.

**Weaknesses:**

- Some important methodological details remain unclear, particularly in the description of the model architecture and loss function.
- The working definition of "interpretable" in the context of this study is not explicitly stated.
- The choice of a threshold (e.g., severity score of 1) is not sufficiently justified with statistical evidence.
- The comparison of computational times with MRIQC is only briefly mentioned in the discussion rather than presented in the methods with quantitative details.

**Detailed Comments:**

- The paper would benefit from a clearer definition of interpretability in this context. Is it related to explainability in deep learning models? If so, how?
- A visual representation of the model architecture would aid comprehension. Additionally, details on what constitutes a "custom convolutional block" should be clarified.
- The variation of artifact severity scores across slices should be reported, given that averaging is performed across middle slices.
- Explicit equations for the loss function, particularly for the L2 loss component, should be included to avoid ambiguity regarding how it encourages low scores for clean images.
- The justification for using a severity threshold of 1 should be supported by quantitative evidence, such as statistical analysis or example distributions.
- The discussion on computational efficiency should be moved to the methods section and supplemented with quantitative comparisons to MRIQC.
- The percentage of datasets with severity scores above 1 should be reported, as this provides useful context for the effectiveness of the method.

**Justification Of The Final Rating:**

Overall the idea is interesting and the results are reasonable. I think the literature review improved the manuscript. On the downside I think the technical details are not fully transparent and a number of comments regarding technical choices and their motivation do not seem fully resolved. So in summary I remain undecided about this manuscript.

**Justification Of The Preliminary Rating:**

Overall the paper addresses a relevant questions. The results seem promising. On the downside, the technical details (which are important to MIDL I think) are missing or not clear enough and some results could be more quantitative. So I am a bit undecided.

**Questions To Address In The Rebuttal:**

1. How do you define "interpretable" in this study, and how does it relate to deep learning interpretability?
2. Could you provide a diagram or clearer description of the model architecture?
3. What specific customizations are applied to the convolutional blocks?
4. How much do artifact severity scores vary across slices before averaging?
5. Could you include explicit equations for the loss function, particularly explaining how L2 loss encourages low scores for clean images?
6. What evidence supports the choice of a severity threshold of 1? Can you provide statistical justification?
7. Why does the dataset restriction lead to testing only one artifact severity level?
8. Can you provide a quantitative comparison of computation times with MRIQC?
9. What percentage of datasets in your experiments had severity scores above 1?

**Special Issue:**

No

---

> ### Author Response · Authors · 2025-03-07
>
> R3/8Z4Q:
>
> In this study, "interpretable" refers to the ability of our model to provide artifact severity scores that are meaningful, continuous, and actionable for QA in MR imaging. Unlike traditional QC metrics that require expert-defined thresholds, our method assigns severity scores that correlate with the visibility and impact of artifacts on image quality. These scores can be used to set thresholds for downstream processing or correction strategies, making our approach more practical for large-scale studies. From a deep learning interpretability perspective, our model does not rely on black-box classification but instead learns a structured embedding space where image quality is ranked in a continuous manner.
>
> *A model architecture diagram was not added due to space, but additional details were added to our model architecture section.
>
> *We comment on the variation of severity scores across slices before averaging.
>
> *We added explicit equations for the loss functions and additional details explaining how the L2 loss encourages low scores for clean images.
>
> The justification for the severity threshold of 1 is based on manual inspection of our test dataset. We do not have statistical justification of it, but this would be a great future validation of the model.
> In our simulated test data, we only explored one artifact at a time. It would be useful in future work to explore images with more than one artifact (i.e., motion and anisotropy)
>
> *We added computational differences between MRIQC and our method in the methods section.
>
> *We reported on the percentage of the private dataset with a score of over 1. This high percentage was expected since most of the data is 2D acquired with large slice thicknesses and slice gaps.

---

### Official Review · Reviewer_NPHf · 2025-02-18

**Confidence:** 4
**Preliminary Rating:** 4
**Recommendation:** Poster
**Final Rating:** 4

**Summary:**

The authors propose a new method for blind quality assurance in magnetic resonance imaging.  The method is validated using simulated artifacts on a public dataset and real artifacts on a private dataset, demonstrating good performance.

**Strengths:**

The method is new and seems to work well in practice, although the methods are pretty straightforward and not especially exciting or technically innovative.  The method is validated using simulated artifacts on a public dataset and real artifacts on a private dataset, demonstrating good performance.

**Weaknesses:**

I think the method is only measuring "apparent" image quality rather than actual image quality.  Actual image quality cannot be judged without knowing the true images.  An image that looks realistic may produce a high score on the proposed metric, but is not necessarily accurate.  For example, this metric will likely have a hard time dealing with adversarial attacks based on realistic-looking but ultimately fake images or images that contain hallucinated realistic structures.  This limits the possible applications of this metric.  For example, this metric probably wouldn't be useful for training a network to perform image enhancement, since there may be a big mismatch between the referenceless apparent quality and reference-based actual quality.  This probably requires some disclaimers about using the metric outside of QA applications, or even in QA applications if advanced image enhancement strategies are also being used.

**Detailed Comments:**

See above

**Justification Of The Final Rating:**

The authors made a minor change in response to my comments (one not-very-prominent sentence was added to the discussion).  I would have preferred something more substantial than this.  My overall assessment is the same as my previous assessment, it's reasonable work with good performance, although pretty straightforward.

**Justification Of The Preliminary Rating:**

The method is reasonable, the validation study is reasonable and convincing, and the performance seems to be good.  The main issues are that the new method seems pretty straightforward (not very innovative), and that there is no discussion of the fundamental limitations of blind image quality metrics.

**Questions To Address In The Rebuttal:**

Quality metrics can be used for various applications, but including some caveats about the dangers of blind image quality metrics would help prevent this metric from being misused.

---

> ### Author Response · Authors · 2025-03-07
>
> R2/NPHf:
>
> *In the discussion we emphasized that our approach measures apparent image quality rather than anatomical realness.

---

### Official Review · Reviewer_pB8z · 2025-02-24

**Confidence:** 4
**Preliminary Rating:** 2
**Final Rating:** 2

**Summary:**

In this work, the authors propose an unsupervised medical image quality assessment framework which assigns an artifact severity score to multi-contrast MR acquisitions. The method involves contrastive learning and synthetic artifact simulations to train a quality assessment model without using labels. The approach aims to address limitations in current quality assessment methods as computational inefficiencies and limited applicability across various image modalities.

**Strengths:**

* Unsupervised learning – The model does not require any labels during training
* Modality-agnostic IQA – The approach is compatible with different brain imaging modalities
* Validation on a private clinical dataset
* Code will be made publicly available

**Weaknesses:**

* Limited related work section – More recent studies on medical image quality assessment should be cited and examined
* The model and experiments need further clarification – Some parts are unclear, requiring assumptions to understand the content

**Detailed Comments:**

* The idea of unifying different types of artifacts into a single severity metric is both practical and compelling. However, I am uncertain whether this framework could be extended to motion correction tasks, as leveraging artifact-specific dynamics could be beneficial in such cases.

* Additionally, I remain unconvinced about the necessity of an unsupervised approach for this problem. While the authors do not explicitly use artifact labels, they still rely on synthetic artifact simulations, which suggests that a weakly supervised or self-supervised approach could have been viable. Furthermore, no prior works are cited to justify why an unsupervised method is particularly well-suited for this task. The introduction should discuss related works and, if possible, include them as comparative baselines in the experiments. Specifically, I recommend referencing:
1. "Brain MRI Artifact Detection and Correction Using Convolutional Neural Networks" – Oksuz, Computer Methods and Programs in Biomedicine, February 2021
2. "Automatic Brain MRI Motion Artifact Detection Based on End-to-End Deep Learning is Similarly Effective as Traditional Machine Learning Trained on Image Quality Metrics" – Vakli et al., Medical Image Analysis, August 2023
3. "Automated MRI Quality Assessment of Brain T1-weighted MRI in Clinical Data Warehouses: A Transfer Learning Approach Relying on Artifact Simulation" – Loizillon et al., MELBA, June 2024

* What is the input dimensionality of your model? Does it require resizing to a fixed dimension?

* Could you mathematically describe how the model shown in Fig. 1 is trained? Conventionally, triplet loss is applied to embedding vectors rather than scalar values. Have you constrained the embedding dimensionality to 1? Clarification on this aspect would be helpful.

* The motivation behind using both L2-norm and triplet loss simultaneously is unclear. Since artifact severity scores are already assigned, why not simply regress the values and compare predictions objectively using metrics like mean absolute error?

* The paper lacks direct comparisons with other deep learning architectures. Have you considered evaluating pre-trained networks or alternative IQA models?

* Have you analyzed the impact of your composite loss function components individually? For instance, does including L2-loss significantly improve performance, or would a pure triplet loss function suffice? Introducing hard negative mining might also enhance robustness.

* Could you report the correlation coefficient between the severity scores and the quality metrics listed in Table 2?

* When generating artifacts with a specific severity score, do you directly set parameters (e.g., anisotropy scale should be set to 2.8 to obtain a severity score of 0.6, right)?

* Have you tested different parameter configurations for each artifact type? Does your model output degrade continuously as artifact severity increases, or do certain artifacts exhibit non-linear behavior?

* Does the validation on the private clinical dataset include a reader study for verification?

* The paper mentions preprocessing steps required by both MRIQC and the proposed method. Do these steps overlap, or does your approach omit certain procedures? A breakdown of the preprocessing pipeline and the time required per volume would be valuable.

**Justification Of The Final Rating:**

I thank the authors for their efforts during the rebuttal process. While the idea is interesting and the authors have partially addressed our questions, I still believe the paper lacks important details regarding reproducibility and motivation for various aspects. Additionally, the results should include more recent comparative methods, and some critical references are missing. For example:

Tian et al., "HyNet: Learning Local Descriptor with Hybrid Similarity Measure and Triplet Loss," NeurIPS 2020

**Justification Of The Preliminary Rating:**

This paper presents an interesting idea but lacks proper justification, strong comparisons, and methodological clarity. The choice of an unsupervised approach is not convincingly argued, especially since artifact labels are available in simulation. Additionally, the evaluation is weak, as it only compares with MRIQC and ignores recent deep learning-based methods, making it difficult to assess the real impact of the proposed approach. These are the reasons behind my weak reject decision

**Questions To Address In The Rebuttal:**

Mentioned in the detailed comments section

---

> ### Author Response · Authors · 2025-03-07
>
> R1/pB8z:
>
> We acknowledge that this work could be extended to correction tasks, but the purpose of this work is artifact detection. Correction of the artifact is beyond the scope of our paper. We could extend this paper to help guide correction.
>
> *We expanded the related work section to include additional studies on medical image quality assessment (Oksuz et al., 2021; Vakli et al., 2023; Loizillon et al., 2024).
>
> *We justified our unsupervised approach, highlighting its ability to scale and its independence from subjective manual labels.
>
> *The model does not require a fixed size. The model rescales the input to the correct size (224x224). We added clarification for the training process, including a mathematical description of triplet loss, an explanation of why we use the L2 loss, and how our loss formulation inherently incorporates hard negative mining.
>
> Foundational models are a promising direction to go but in this work our focus was on the triplet loss and unsupervised framework.
> We have trained a model without the L2 loss and margin in the triplet loss. This led to uninterpretable model output scores. For example, a higher score did not indicate lower or higher image quality.
>
> *A new correlation analysis between our scores and MRIQC metrics has been included in Table 2
>
> A specific severity score does correspond directly to set parameters as described in Table 1. This does allow for continuous and linear severity scoring.
>
> We did not include a reader study but this could be done in the future for further validation.
>
> *We added computational differences between MRIQC and our method. Our approach omits all preprocessing on the NIFTI images. Our approach only requires a NIFTI image.

---

### Author Rebuttal · Authors · 2025-03-07

**Rebuttal:**

We appreciate reviewers’ valuable and constructive suggestions. Below, we address each major concern and outline revisions accordingly. More detailed concerns are added as official comments to each reviewer. Changes to the paper in response to reviewer comments are marked with an asterisk * in this rebuttal and clearly marked in the updated manuscript.

R1/pB8z:
We acknowledge that this work could be extended to correction tasks, but the purpose of this work is artifact detection.
*We expanded the related work section to include additional studies
*We justified our unsupervised approach, highlighting its ability to scale and its independence from subjective manual labels.
*We added clarification for the model input size, training process, including a mathematical description of triplet loss, an explanation of why we use the L2 loss, and how our loss formulation inherently incorporates hard negative mining.
*A new correlation analysis between our scores and MRIQC metrics has been included in Tab2
*We added computational differences. Our approach omits all preprocessing on the NIFTI images.

R2/NPHf:
*In the discussion we emphasized that our approach measures apparent image quality rather than anatomical realness.

R3/8Z4Q:
"Interpretable" refers to the ability of our model to provide artifact severity scores that are meaningful, continuous, and actionable for QA in MR imaging. From a deep learning interpretability perspective, our model does not rely on black-box classification but instead learns a structured embedding space where image quality is ranked in a continuous manner.
*A model architecture diagram was not added due to space, but additional details were added to our model architecture section.
*We comment on the variation of severity scores across slices.
*We added equations for the loss functions and additional details.
The justification for the severity threshold of 1 is based on manual inspection of our test dataset.
In our simulated test data, we only explored one artifact at a time. It would be useful in future work to explore images with more than one artifact (i.e., motion and anisotropy)
*We added computational differences  in the methods section.
*We reported on the percentage of the private dataset with a score of over 1.

We believe these revisions strengthen the paper significantly and address the concerns to the best of our ability. Thank you again for your insightful feedback.
Best regards,
Savannah Hays & Co-authors

**Supporting Material:**

/attachment/335066a3b48abcecd201b919f985346d8ca01df2.pdf

---

### Meta-Review · Area_Chair_mqfd · 2025-03-17

**Recommendation:** Accept (Poster)
**Confidence:** 5

**Metareview:**

The reviewers had several concerns with the original submission, many of which have been successfully addressed during the rebuttal stage. Even though the authors have tried to include more details on their method within the space constraints, there are still issues with the clarity of the description of their algorithm. There are also issues with the limited number of competing methods, and some minor concerns about limited novelty and lack of robustness to adversarial attacks. Despite these outstanding issues, all reviewers agree that the idea is interesting, which is why I am recommending acceptance as a poster (which will hopefully generate good discussion during the conference).